# Elite Cacao Clonal Cultivars with Diverse Genetic Structure, High Potential of Production, and Good Organoleptic Quality Are Helping to Rebuild the Cocoa Industry in Brazil

**DOI:** 10.3390/ijms26073386

**Published:** 2025-04-04

**Authors:** Lívia Souza Freitas, Gonçalo Santos Silva, Ivanildes Conceição dos Santos, Adriana C. Reis Ferreira, Laysa Evelin Silva Santos, Pathmanthan Umaharan, Lambert A. Motilal, Juan Calle-Bellido, Dapeng Zhang, Ronan Xavier Corrêa, Dário Ahnert

**Affiliations:** 1Departamento de Ciências Biológicas, Universidade Estadual de Santa Cruz, Ilhéus 45662-900, BA, Brazil; livia_freitass@hotmail.com (L.S.F.); darioahnert1@gmail.com (D.A.); 2Centro de Inovação do Cacau, Ilhéus 45662-900, BA, Brazil; ivanildessant@gmail.com (I.C.d.S.); adrianabiocau@gmail.com (A.C.R.F.); laysaevelin@outlook.com.br (L.E.S.S.); 3Cocoa Research Unit, The University of the West Indies, Saint Agustine 311331, Trinidad and Tobago; pathmanathan.umaharan@sta.uwi.edu (P.U.); lambert.motilal@sta.uwi.edu (L.A.M.); 4Mondelez International, Inc., East Hanover, NJ 07936, USA; juan.calle.bellido@mdlz.com; 5Sustainable Perennial Crops Laboratory, United States Department of Agriculture, Agriculture Research Service, Beltsville, MD 20705, USA; dapeng.zhang@usda.gov; 6Centro de Biotecnologia e Genética, Departamento de Ciências Biológicas, Universidade Estadual de Santa Cruz, Ilhéus 45662-900, BA, Brazil

**Keywords:** molecular breeding, cacao clones, production, cocoa bean quality, organoleptic, genetic ancestry, SNP

## Abstract

In the Americas’ leading cocoa-producing countries, more productive clonal cultivars than traditional biclonal hybrids have been created. In Brazil, several disease-resistant and self-compatible clones such as PS 1319, FA 13, and SJ 02 have been selected on producer farms. The CCN 51 clone from Ecuador is also significant in Brazil. This study aimed to analyze these clones concerning their genetic structures using single-nucleotide polymorphisms, productive potential, disease resistance, and the physico-chemical and organoleptic characteristics of the beans. Clone SJ 02 has ancestry from Contamana (40.7%), Iquitos (34.5%), and Amelonado (23.5%). PS 1319 is primarily Amelonado (67.9%), with Criollo (15.7%) and Contamana (15.6%). FA 13 mainly consists of Amelonado (53.5%) and Iquitos (44.1%). Local cultivars of Bahia are mostly Amelonado, with 99.8% in Comum and Parazinho, 97.4% in Maranhão, and 95.5% in Pará. PS 1319, SJ 02, and FA 13 clones were significantly more productive than CCN 51 but did not differ in disease resistance levels. Significant differences were noted among the cultivars in physicochemical traits (fat, caffeine, and theobromine content). Sensorially, SJ 02 outperformed the other cultivars and was comparable to the reference clone BN 34. The findings indicate that Brazil’s elite clones, derived from complex crosses involving Amelonado, Contamana, Iquitos, and Criollo groups, are productive, resistant, and exhibit favorable physico-chemical and organoleptic qualities, making them valuable for future clonal breeding programs.

## 1. Introduction

Over the past two decades, genetic improvement for cocoa in Brazil has concentrated on developing productive, disease-resistant, and self-compatible clones [1,2]. Currently, clonal cultivars are the primary planting material for establishing new, technologically advanced, and highly productive cocoa plantations in the country. These cultivations occur both in traditional growing regions with humid climates and in the Brazilian Cerrado and Semi-Arid regions. In the Cerrado and Semi-Arid areas, cocoa cultivation is characterized by mechanized planting on flat terrain, with irrigation and full sun exposure.

Several clonal cocoa cultivars have been selected in Brazil, and others are under the selection process [1,3]. Notably, the cultivars PS 1319 (Fazenda Porto Seguro—Bahia), SJ 02 (Fazenda São José—Bahia), and FA 13 (Fazenda Argolo—Bahia) are among the most extensively utilized [1,4,5]. These cultivars were identified through mass selection on producing farms in the state of Bahia. In addition, the cultivar CCN 51, due to its particular significance, was introduced into Brazil from Ecuador between the 1980s and 1990s and was recommended by the CEPEC (Cocoa Research Center) from Comissão Executiva do Plano da Lavoura Cacaueira (CEPLAC) [1,6]. These four clones have revitalized Brazil’s cocoa cultivation in the last decade.

PS 1319, SJ 02, and FA 13 cultivars are self-compatible and disease-resistant. They exhibit high production potential, good organoleptic quality, and a more upright architecture, making them highly favored by producers. They are propagated through the grafting of plagiotropic shoots onto seminal rootstocks or grafted onto basal shoots of older plants. PS 1319 and SJ 02 have red pods when immature, with an average mass of 998.8 g and 1217.4 g, respectively. Both cultivars possess beans with an average mass of 1.5 g and have a pod index of 20. In contrast, when immature, FA 13 has green pods with an average mass of 893.3 g, smaller beans averaging 1.1 g, and a pod index of 22 [1,3].

The pursuit of effective strategies for the genetic improvement of cocoa has intensified in response to challenges such as the progression of diseases, climate change, reduction in field labor, aging plantations, the need for mechanization, and the enhancement of bean quality. A global deficit of approximately 400,000 tons of cocoa beans is anticipated for the 2023/2024 crop year, causing significant disruptions to the supply chain [7]. This decline in production for the third consecutive year reflects a series of technical issues that necessitate concerted efforts within the supply chain to improve the global production system.

Population improvement through the exploitation of heterosis is considered one of the most effective strategies for cocoa in traditional breeding [2]. This approach can be operationalized using the reciprocal recurrent selection method, which involves improving parent genotypes over selection cycles to develop new cultivars [2,6]. Furthermore, to accelerate conventional breeding efforts, molecular biology tools such as gene editing, genomics, proteomics, molecular markers, and bioinformatics are being utilized [8,9,10,11,12,13]. The use of molecular markers, including Simple Sequence Repeats (SSRs) and single-nucleotide polymorphisms (SNPs), enables precise and comprehensive genetic analysis of the species [14,15,16].

Disease-resistant, self-compatible, and productive clones selected in Brazil on producer farms, such as PS 1319, FA 13, and SJ 02 [3], monilia-resistant clones selected from germplasm banks in Costa Rica [17,18], CCN clones selected in Ecuador [19,20,21], and many others currently in selection and evaluation stages in producing countries represent a significant source of germplasm. These clones, which are advanced due to breeding processes and carry favorable alleles for cultivation, require further study. They offer the potential to be utilized more efficiently in breeding programs than accessions that have not yet been selected [2].

This study aimed to characterize three clonal cultivars of significant importance to Brazil (FA 13, PS 1319, and SJ 02) in terms of their genetic ancestry, phylogeny, production potential, resistance, organoleptic, and physicochemical qualities. The scope of the study encompassed both genetic and agronomic evaluations, providing cocoa breeders with valuable insights to leverage these elite clones for improving cacao productivity and quality in Brazil and other cacao-producing regions.

## 2. Results

### 2.1. Genetic Structure and Ancestry

A total of 192 SNP markers were utilized, of which 187 were informative. The genetic structure analysis allowed us to understand the ancestry of the clones and their relationship with the genetic groups of cocoa described by Motamayor [8] (Figure 1). The four local cultivars from Bahia, the three clonal cultivars, and the ten reference accessions from the International Cocoa Genebank Trinidad (ICGT) showed differences in genetic composition (Figure 2).

The genetic structure analysis of Brazilian genotypes and reference accessions from the ICGT revealed that the clonal cultivars (PS 1319, FA 13) and the local cultivars (Comum, Pará, Parazinho, and Maranhão) exhibited the Amelonado genetic group with the highest representation in their genetic structure (Figure 2). Although Amelonado is also present in clone SJ 02, this cultivar predominantly comprises the Contamana and Iquitos groups. The FA 13 clone is almost entirely composed of a combination of Amelonado and Iquitos groups. The local cultivars from Bahia showed over 95% Amelonado, but a complex mixture of multiple genetic groups was also detected in Pará and Maranhão (Table 1).

### 2.2. Genetic Relationship

The phylogenetic results placed the clonal cultivars FA 13, PS 1319, and SJ 02 near the Amelonado group (Figure 3). The local cultivars from Bahia were positioned within the Amelonado group. Therefore, the phylogenetic results were consistent with the genetic structure obtained from the Bayesian analysis (Figure 2).

The reference cultivars from the ICGT, elite clonal cultivars, and local cultivars from Bahia were clustered within or close to the Amelonado group (Figure 3). The Brazilian clones formed a group near the Amelonado, while the local cultivars were included within this genetic group.

The dendrogram in Figure 4 illustrates the formation of three major clades and their subgroups. One clade includes three local cultivars from Bahia (Pará, Parazinho, and Comum) and SIC 5, an accession from Amelonado in Bahia. The second group comprises the local Bahia cultivar Maranhão and several other reference Amelonado accessions. The third group includes Amelonado accessions, ICS cultivars, Brazilian clonal cultivars, M 8, and CCN 51.

The cultivar PS 1319 clustered near CCN 51, a sister clade to the ICS accessions. In contrast, the accessions SJ 02 and FA 13 formed a separate clade, which was a sister clade to the clade with PS 1319, CCN 51, and ICS accessions.

### 2.3. Agronomic Characteristics of Elite Cacao Clones in Three Production Systems

The analysis of variance revealed statistically significant differences among the clonal cultivars for the number of healthy pods per plant across the three production systems: full sun, thin cabruca, and dense cabruca (Table 2). Additionally, significant differences were observed in productivity (kg plant^−1^) in the full sun and dense cabruca systems (Table 2).

Under dense cabruca conditions, the clonal cultivar FA 13 produced the highest number of healthy pods (pods plant^−1^) and the greatest production of dry beans (kg plant^−1^) with values 226 and 115% higher, respectively, compared to CCN 51 (Table 2).

The clonal cultivars did not show statistically significant differences (*p* < 0.05) in the percentages of pods infected by witches’ broom per plant, pod rot per plant, and damaged pods per plant (mummified and others) across the three production systems: full sun, thin cabruca, and dense cabruca, respectively (Table 3).

Although no significant differences were observed between clonal cultivars for the percentage of witches’ broom and black pod rot when cultivated under full sun conditions, the clonal cultivars FA 13, PS 1319, and SJ 02 recorded percentages of pods with witches’ broom of 144%, 172%, and 165% higher, respectively, compared to CCN 51.

In plants cultivated under the thin cabruca system, the clonal cultivar CCN 51 exhibited more witches’ broom and black pod rot in pods. Conversely, in the dense cabruca production system, FA 13, PS 1319, and SJ 02 recorded 9, 28, and 33% higher values for pods with witches’ broom and lower values for black pod rot, respectively, compared to CCN 51.

### 2.4. Physical, Chemical, and Sensory Characteristics of the Cocoa Beans from Clonal Cultivars

The clonal cultivars exhibited differences in physical, chemical, and sensory characteristics (Table 4). Among the physical characteristics, BN 34 and CCN 51 demonstrated the highest average bean mass, resulting in fewer beans per 100 g. Additionally, the pH at 25 °C was lower for all clonal cultivars compared to BN 34, with values of 6.6% for CCN 51 compared to BN 34. Regarding the chemical properties, the highest values for the assessed attributes were observed in BN 34 and FA 13, which presented similar purine values. Additionally, FA 13 exhibited caffeine content values that were 26.3% higher compared to BN 34. In contrast, Catongo had the lowest caffeine content, 63.2% lower than BN 34, and stood out with theobromine levels 2.8% higher than BN 34. The fat content of cocoa beans varied between clonal cultivars. Catongo, FA 13, and SJ 02 cultivars exhibited fat values similar to BN 34. In contrast, the cultivar PS 1319 had a fat content of 6.9% lower compared to BN 34.

The clonal cultivars showed variation in total polyphenol content in the beans, with values ranging from 15.7% to 67.7% lower for Catongo and SJ 02, respectively, compared to BN 34, which recorded the highest value. Additionally, Catongo exhibited the highest theobromine/caffeine ratio (T/C), 183.2% higher, while FA 13 had a T/C ratio of 24.1% lower compared to BN 34.

Regarding organoleptic properties, SJ 02 received the highest scores for sweetness, with values similar to BN 34. The highest ratings for cocoa flavor and bitterness were observed in the cultivars CCN 51, FA 13, and PS 1319, with scores of 14.9, 19.0, and 21.5%, respectively, for cocoa flavor and 12.4, 4.1, and 9.1%, respectively, for bitterness, compared to BN 34. Conversely, these cultivars received the lowest scores for sweetness.

For auxiliary attributes (fruity, floral, nutty, spicy, and woody), SJ 02 and FA 13 stood out for their fruity notes, with values of 10.2 and 5.2% for fresh fruits and 35.4 and 18.5% for brown fruits, respectively, the highest compared to BN 34. Additionally, SJ 02 excelled in floral and spicy attributes, with values 19.1% lower and 8.6% higher, respectively, compared to BN 34. Notably, BN 34 distinguished itself with a high rating of 2.71 for the woody attribute.

Both for the chemical and organoleptic attributes, the first two principal components in the principal component analysis (PCA) explained cumulatively over 80% of the total variation (Table 5). For chemical attributes, PC 1 and PC 2 accounted for 53.63% and 28.35% of the total variation, respectively. In PC 1, the attributes of theobromine, polyphenols, and T/C had the greatest contribution. For PC 2, caffeine had the greatest contribution. PC 1 explained 66.41% of the total variation for organoleptic attributes, with bitterness, sweetness, fruitiness, and spiciness being the major contributors. PC 2 accounted for 25.45% of the total variation, with cocoa flavor and acidity being the major contributors.

In the PCA of chemical attributes (Figure 5A), the clonal cultivar BN 34 and the local cultivar Catongo were primarily distinguished by high concentrations of polyphenols and theobromine. Conversely, SJ 02 and PS 1319 exhibited lower values for these attributes. Additionally, the clonal cultivar FA 13 was notably characterized by high caffeine content and a low T/C ratio, while CATONGO displayed low caffeine content and a high T/C ratio.

In PC 1 for organoleptic attributes (Figure 5B), SJ 02 and BN 34 were distinctly grouped apart from PS 1319 and CCN 51. This separation was primarily due to higher fruitiness, spiciness, and sweetness scores combined with lower bitterness scores. Additionally, SJ 02 and FA 13 exhibited higher acidity and cocoa flavor scores, distinguishing them from BN 34.

## 3. Discussion

### 3.1. Genetic Structure and Diversity

Over the past two decades, cacao-producing countries in the Americas have increasingly focused on the production of clonal cacao cultivars as planting material due to their superior agronomic potential compared to traditional interclonal hybrid crosses [2]. In Brazil, clonal cultivars such as FA 13, PS 1319, SJ 02, and CCN 51 are widely cultivated by farmers, representing optimal choices for cultivation in both humid regions (Amazon and Bahia) and dry regions (Cerrado and Semi-Arid). Currently, new cultivars are under selection, with some already being released and tested by producers, suggesting that farmers will shortly have access to a wide range of clonal cultivars [1,2].

The genetic structure analysis conducted in this study revealed that the SJ 02 cultivar exhibits a predominance of DNA from the Contamana (40.7%), Iquitos (34.5%), and Amelonado (23.5%) groups. The PS 1319 cultivar predominantly contains Amelonado (67.9%), Criollo (15.7%), and Contamana (15.6%) DNA. The FA 13 cultivar primarily comprises Amelonado (53.5%) and Iquitos (44.1%) DNA. The CCN 51 cultivar predominantly includes DNA from the Iquitos, Criollo, and Amelonado groups (Figure 2), in agreement with Boza [23]. These findings suggest that these cultivars result from complex crosses involving germplasm from three or more genetic groups.

The FA 13, PS 1319, and SJ 02 clones originated from hybrids distributed by CEPLAC to farmers in Bahia, Brazil, and were selected through mass selection on producer farms [2,3]. In Bahia, the following accessions were primarily used as parents for the production of biclonal hybrids released to farmers: ICS 1, ICS 6, ICS 8, UF 613, UF 168, UF 667 (Trinitarios); IMC 67, PA 30, PA 150, SCA 6, SCA 12 (Upper Amazon Forasteros); TSA 644 (SCA 6 × IMC 67); and SIAL 70, SIAL 169, SIAL 325, SIAL 505, SIC 17, SIC 19, SIC 328, SIC 329, and SIC 813 (Lower Amazon Forasteros) [24].

For hybrid production, crosses were made between Upper Amazon Forastero × Lower Amazon Forastero, Upper Amazon Forastero × Trinitario, and Trinitario × Lower Amazon Forastero [2]. The clonal cultivars PS 1319 and SJ 02 exhibit red pods when immature, suggesting that they inherited this trait from the Trinitario clones ICS 1, UF 613, or UF 667, as these are the only clonal accessions listed above with red pods. The red color of immature pods is a dominant trait over the green color [25]. The potential ancestry of PS 1319 and SJ 02 from the ICS 1, UF 667, and UF 613 clones is further supported by multivariate analysis results, which grouped the reference Trinitarios of this study within the same subgroup as these clones in the dendrogram (Figure 4).

The clones SJ 02, PS 1319, and FA 13 were selected on producer farms for their resistance to witches’ broom (*Moniliophtora perniciosa*), high productivity, and desirable pod and seed characteristics [2]. The Upper Amazon genitors SCA 6, SCA 12, PA 150, PA 30, and IMC 67 were employed as sources of pest resistance and other agronomic traits, in addition to providing hybrid vigor for hybrid production in Brazil [2]. Specifically, the SCAs, PA 150, and IMC 67 were utilized as sources of resistance to *M. perniciosa*. The Upper Amazon Forastero IMC 67 belongs to the Iquitos group, PA 30 and PA 150 to the Marañon group, and SCA 6 and SCA 12 to the Contamana group. Therefore, SJ 02 likely had IMC 67 and SCA as its principal progenitors, given that its genome contains 44.7% Contamana and 34.5% Iquitos DNA (Figure 2). Following the same logic, the PS 1319 clone likely had SCAs as one of its main progenitors, while the FA 13 clone likely had IMC 67 as one of its principal progenitors.

This study is the first to reveal that the local cultivars Pará and Maranhão, originating from the local cacao population in Bahia, are not pure Amelonado. Instead, they contain a small amount of genome from the Marañon group and other genetic groups (Figure 2, Table 1). Before this study, it was believed that the local cultivars in Bahia were composed entirely of Amelonado group genomes, with the existing genetic variability exclusively attributed to this group [25,26]. In contrast, Comum and Parazinho’s local cultivars analyzed in this study possess only Amelonado group DNA.

The Cacau Comum arrived in Bahia in 1746 from the Amazon region, and this material likely introduced genetic material from the Amelonado group. Over the subsequent 100 years, several additional introductions occurred, leading to the development of local cultivars such as Maranhão, Pará, and Parazinho [25]. These different local cultivars have been crossing randomly among themselves over multiple generations, resulting in a local population predominantly composed of the Amelonado group genome. However, these varieties also include Marañon and other genetic groups, as demonstrated in this study. This information elucidates the origin of the genetic diversity within the local cacao population of Bahia and is crucial for selecting parent lines in breeding programs.

The accessions SIAL 70, SIAL 169, SIAL 325, SIAL 505, SIC 17, SIC 19, SIC 328, SIC 329, and SIC 813 originated from selections made in Bahia between 1930 and the 1950s. These accessions are potential Amelonado progenitors of the FA 13, SJ 02, and PS 1319 clones. Although Criollo group descendants are in smaller quantities in these elite clonal cultivars (Table 1), their contribution remains significant. Additionally, Trinitario cultivars, such as ICS and UFs, which originated from crosses between Amelonado and Criollo varieties and were selected by agronomists, have played and continue to play a crucial role in the genetic improvement of cacao in Brazil, mainly contributing to seed size and bean quality [3,25].

Finally, it is essential to recognize that in genetic breeding, the quantity and quality of germplasm are crucial; however, the most critical factor is the quality of the germplasm to attend to agronomic demands. Overall, the results of this study underscore the significant role of the Amelonado, Contamana, Iquitos, and Criollo groups in the composition of elite clonal cultivars in Brazil. Another essential aspect to consider is the practical significance of the use of these three elite clonal cultivars as progenitors in breeding programs. Utilizing these elite clonal cultivars in a breeding program in addition to CCN 51 and other good clonal cultivars selected in farmers’ fields in Brazil can facilitate the rapid development of new and improved cultivars. This approach leverages a set of progenitors that have already been genetically improved, thus exhibiting a higher frequency of favorable alleles for cacao cultivation.

### 3.2. Agronomic Characteristics

The elite cultivars FA 13, SJ 02, PS 1319, and CCN 51 are widely cultivated by producer farms in humid regions (Amazon and Bahia) and dry regions (Cerrado and Semi-Arid) of Brazil [27,28]. FA 13, PS 1319, and CCN 51 are easier to prune than SJ 02, both manually and mechanically, and are commonly used in mechanized cultivation systems. Regarding pod index, CCN 51 requires approximately 15 pods to produce 1 kg of dried cocoa beans, SJ 02 and PS 1319 require 20 pods each, and FA 13 requires 22 to 24 pods. To achieve these indices, proper fertilization and management practices are necessary [29,30,31].

This study aimed to indicate the production potential of elite clonal cultivars in Brazil by comparing them to CCN 51, a widely cultivated and high-yielding clone. To assess production potential, pod yield data from clonal cultivars were used for only the year from which production became stable in a multi-year series of assessments, specifically from the sixth to the seventh year of cultivation, when the plants had reached production stability. In the present study, the FA 13 and SJ 02 cultivars outperformed CCN 51 regarding the number of pods and plant productivity. Although these cultivars have a higher pod index, their production is offset by the greater number of pods produced per plant, which was approximately twice as many compared to CCN 51.

The average production of the clones across the three cultivation systems was as follows: FA 13 = 3.3 kg plant^−1^, PS 1319 = 3.0 kg plant^−1^, SJ 02 = 2.6 kg plant^−1^, and CCN 51 = 2.5 kg plant^−1^ (Table 2). Therefore, they have a high potential for production and the difference in productivity among themselves is due to their genetic makeup (Figure 4) and their acclimatization to each location/cultivation system. These results show that the elite clonal cultivars of Brazil have similar or even superior production potential under certain environmental conditions compared to CCN 51. On the other hand, the pod and seed sizes of CCN 51 are larger than those of FA 13, PS 1319, and SJ 02 [3,31], which is consistent with the results for the number of seeds per 100 g reported in this study. However, the seed size of all these clones is suitable for the industry, as observed in the technical recommendations for fermentation specific to each seed size range.

The cultivar PS 1319, over the years of cultivation, has increasingly exhibited a high number of witches’ brooms on floral cushions in humid regions, possibly indicating an adaptation of *M. perniciosa* to this clone. However, with integrated control measures (cultural, biological, and chemical), pod losses and witches’ broom occurrences on floral cushions are reduced, as observed in this study. Farmers continue cultivating PS 1319 despite this issue due to its high productivity. FA 13, SJ 02, and CCN 51 maintain resistance to *M. perniciosa* on floral cushions. The cultivar SJ 02 requires specific environmental conditions for successful cultivation and does not acclimate well to all planting conditions. In contrast, the other cultivars adapt well to almost all environments (Ahnert, D., personal communication).

The clonal cultivars did not show statistically significant differences in the percentages of pods infected by witches’ broom plant^−1^, black pod rot plant^−1^, and damage plant^−1^ (e.g., mummified and others) (Table 3). This indicates that, like CCN 51, they are relatively resistant to diseases when properly managed, as evidenced by the low percentage of infected pods observed across the four clonal cultivars evaluated. Notably, the PS 1319 cultivar, although known to be susceptible to *M. perniciosa*, exhibited a percentage of pods with witches’ broom symptoms similar to that of CCN 51, particularly in full-sun and thin cabruca environments.

### 3.3. Physical, Chemical, and Sensory Characteristics of Cocoa Beans

The beans from the BN 34 cultivar are dense and, in this experiment, exhibited an atypical density greater than that of CCN 51, which is known for having beans with a higher density among elite clonal cultivars [23,32,33,34,35]. Therefore, this result may be related to the particularities of the harvest due to biotic or abiotic factors, as discussed by Almeida & Valle [36].

The physicochemical and sensory analyses of the studied cultivars allowed for the exploration and confirmation of their potential for fine chocolate production and the presence of beneficial chemical compounds for the industry. The evaluated cultivars exhibited distinct chemical and sensory profiles. The FA 13 cultivar stood out for its purines, caffeine, and fat levels, with values comparable to those of BN 34. The SJ 02 and Catongo cultivars also showed fat content similar to BN 34. In contrast, SJ 02 exhibited a lower polyphenol content, whereas BN 34 demonstrated a high level of this compound.

In the present study, the CCN 51, FA 13, and PS 1319 cultivars exhibited higher bitterness scores than BN 34. In contrast, SJ 02 displayed sweetness, fruity, floral, and spicy notes similar to or greater than BN 34. Additionally, FA 13 also showed fruity notes comparable to BN 34. While CCN 51 and PS 1319 are not preferred for fine chocolate production due to their higher bitterness and astringency, they are widely used for bulk cacao production and cacao derivatives [37,38]. These cultivars are recognized as some of the best for cacao nectar extraction, sweets, and jams due to the properties of their pulp [33,39]. For fine cacao production, these unfavorable characteristics must be managed by producers during the cacao processing stages (harvest, fermentation, and drying) [40,41,42,43].

Cocoa post-harvest is essential for the quality of the beans. During the fermentation and drying stages of the beans, several chemical compounds such as methylxanthines, polyphenols, proteins and carbohydrates present in the beans participate in the formation of specific flavors present in the cocoa beans. Flavor is one of the most important and decisive quality attributes for the acceptability of cocoa products, such as chocolate [44].

In addition to characterization, the two principal component analyses provided a deeper understanding of how chemical and organoleptic attributes contributed to the grouping and distinction of the studied cultivars. It was observed that chemical attributes such as theobromine, polyphenols, purines, and the theobromine/caffeine ratio and organoleptic attributes including bitterness, sweetness, fresh fruit, and spice were significant in differentiating the cultivars. The cultivars BN 34 and SJ 02 are clearly identified and grouped separately from the other cultivars, primarily due to their similar sensory profiles. BN 34 is frequently used as a reference due to its recognition in international competitions [45], while SJ 02, a finalist in national competitions over the past three years, is distinguished by its superior scores in sweetness, fruitiness, and spiciness, even when compared to BN 34 [44]. Analyzing fine cacao clones from Brazil, such as BN 34 and SJ 02, highlights their significance for the premium cacao industry. This sensory superiority underscores the importance of sensory characterization work in identifying and selecting clones with the potential for fine cacao production.

The concept of fine cacao is related to the absence of off-flavors, the balance of key attributes with low bitterness, acidity, and astringency, and prominent auxiliary attributes that stand out in the sample and provide a signature and identity for the chocolate [46,47]. Beans from cacao trees with fruity, floral, or spicy flavors indicate a genetic potential for fine cacao [44,46]. Since flavor is also influenced by intrinsic factors such as plant genetics, genetic profiles can be selected along with cultivation conditions (sun, rainfall, soil nutrients). Additionally, controllable factors contribute to the quality and development of cacao flavor and aroma, such as post-harvest seed handling, fermentation, and drying [48,49]. The complexity of cacao’s chemical composition becomes apparent due to its significant changes during the post-harvest process [50].

Sensory characterization studies are a relatively recent but essential approach in cacao research such as those conducted by Aprotosoaie et al. [51] and Deus et al. [52], and are crucial for advancing genetic breeding by providing detailed information on the chemical and organoleptic attributes of different clones. This information is essential for breeders and producers seeking to enhance cacao quality. Additionally, studies such as those by Sukha [53] investigate the relationship between chemical compounds, such as polyphenols, and sensory characteristics, highlighting the complexity of cacao flavor profiles.

Another important aspect is the influence of volatile compounds on the perception of flavor and aroma in cacao, as explored by Aprotosoaie et al. [51], who examined the chemical composition of cacao and its implications for health and sensory quality. These studies emphasize the importance of a multidisciplinary approach in cacao research, integrating chemistry, genetics, and sensory aspects to develop high-quality products. Finally, this study highlights the SJ 02 cultivar as having significant potential for fine cacao, due to its sensory attributes showing low levels of bitterness and astringency and high scores for auxiliary attributes such as fruitiness and spiciness. These characteristics give this clone a unique and appealing flavor for the high-quality chocolate market, adding greater value to the final product.

## 4. Materials and Methods

### 4.1. Plant Material and Genotyping

Leaf disc samples from three elite cacao clones (FA 13, PS 1319, and SJ 02), which are the focus of our study (Appendix A), were collected. Additionally, leaf disc samples were collected from four local cultivars in Bahia: Cacau Comum, Pará, Parazinho, and Maranhão (Appendix A). These materials have been extensively used as parents in breeding programs for obtaining cacao hybrids at Comissão Executiva do Plano da Lavoura Cacaueira—CEPLAC [3]. The samples were collected at Fazenda Boa Sorte, in the municipality of Uruçuca, Bahia, Brazil (14°37′04.0″ S 39°15′39.4″ W). After collection, the samples were sent to LGC Genomics (Hoddesdon, UK) for DNA extraction and genotyping using KASP (Kompetitive Allele Specific PCR) for SNP detection. Thousands of SNP loci were identified from cDNA sequences of expressed genes from different plant organs across the ten chromosomes of the cacao genome [54,55]. An informative panel of 192 discriminant SNPs was developed at the Cocoa Research Centre at the University of the West Indies [56,57]. The selection of this SNP panel was based on the high level of polymorphisms and its ability to distinguish closely related accessions [57,58].

This study also included genotypic data from accessions belonging to the ten international genetic groups defined by Motamayor [8]: Amelonado (16 accessions), Contamana (8 accessions), Criollo (12 accessions), Curaray (14 accessions), Guiana (20 accessions), Iquitos (20 accessions), Marañon (20 accessions), Nacional (20 accessions), Nanay (20 accessions), and Purús (5 accessions). Reference accessions were obtained from the International Cocoa Genebank Trinidad (ICGT).

The study also included accessions deemed necessary by the authors for understanding the genetic base of elite clones and the landraces: CCN 51, ICS 1, ICS 40, ICS 60, ICS 95, M 8, MATINA 1/6, REDAMEL 1/27, REDAMEL 1/30, and REDAMEL 1/31. CCN 51 is a high-yielding variety from Ecuador and is currently cultivated in several cacao-producing countries [32,34]. The four Imperial College Selection (ICS) accessions have been extensively used in breeding programs and commercial plantations due to their favorable agronomic characteristics. The three REDAMEL accessions, or “RED AMELONADO”, were additional references for Amelonado. M 8 is an accession from the second generation of a spontaneous cacao population discovered by G. Stahel in 1923 in the Amazon Region at Mamaboen Creek, a tributary of the Coppename River in southern Suriname [25]. MATINA 1/6 is an Amelonado accession from Costa Rica [25].

### 4.2. Genetic Structure and Ancestry

The genetic structure analysis in this study was performed using the software STRUCTURE 2.3.4 [22]. The software uses a Bayesian approach to assign individuals to a predetermined number of populations (K). Thus, the STRUCTURE software estimates the proportion of alleles from each of these populations in each individual’s genome. Additionally, the program calculates the likelihood of different tested models, considering various population configurations (K).

Two bar plot graphs were constructed with the generated data: one showing the ancestral background of all the genetic materials used in this study and another displaying the genetic structure of Brazilian accessions (FA 13, PS 1319, SJ 02, Cacau Comum, Pará, Parazinho, and Maranhão) alongside reference accessions from the Trinidad germplasm to understand the genetic base of elite clones (CCN 51, ICS 1, ICS 40, ICS 60, ICS 95, M 8, MATINA 1/6, REDAMEL 1/27, REDAMEL 1/30, REDAMEL 1/31).

The model used in the analysis of this study assumes the possibility of gene flow or admixture between populations (admixture model) with an inferred alpha value and independent allele frequency. The admixture model was used in this study based on the assumption that cultivated cacao cultivars possess mixed ancestry derived from multiple populations. Independent allele frequency assumes that allele frequencies in different populations or groups are independent of each other. A burn-in of 200,000 iterations and 300,000 replications was collected via Markov Chain Monte Carlo (MCMC), with 20 iterations for each value of K. The burn-in length allows the algorithm to reach a stable state before data collection begins and longer burn-in periods ensure that the chain has stabilized. The number of MCMC interactions affects the precision of the parameter estimates and longer MCMC runs generally provide more reliable estimates. The evaluation was conducted to test the expected separation of ten genetic groups, with convergence observed in 18 out of 20 independent STRUCTURE runs, indicating strong support for the inferred population structure. Ancestry estimates from 18 “q” files were used to obtain the average contributions from each of the ten genetic clusters.

In this analysis, the value of K ranged from 1 to 10, with ten independent repetitions for each. The variable K represents the number of predefined genetically distinct populations. From these STRUCTURE analyses, it was possible to determine the number of populations that best explain the dataset and, therefore, define this important parameter without prior knowledge of the genetic structure present in the samples.

The highest positive ln P(D) run was selected to represent the ancestral background. A minimum level of 5% was used as evidence for the presence of a genetic group. A minimum level of 95% with no 5% level in any other group was used to establish an exclusive association with a genetic group.

### 4.3. Phylogenetic Analysis

The phylogenetic analysis was performed using DARwin 6.0.16 [59]. The generated phylogenetic tree relates all the accessions in this study to the genetic groups. Additionally, the Amelonado clade figure illustrates the genetic relationship of the seven Brazilian accessions (FA 13, PS 1319, SJ 02, Cacau Comum, Pará, Parazinho, and Maranhão) with the ten reference accessions from ICGT that are positioned within or near it (Amelonado group reference accessions: MAR 14, SIC 5, GC T 998/2, GDL 7, N38 [T38], AMELONADO [VAR], MAR 13, LCT EEN 62/S-4, DOM 24, DOM 18, DOM 21, MAR 11, GS 13, CERRO AZUL 10, EEG 8, M 252 [ICGT]). For allelic data, the program created a simple matching dissimilarity index. Missing data were handled by excluding alleles paired with a minimum of 70% valid variables in bootstrap support from 1000 replicates. The tree construction employed the weighted Neighbor Joining algorithm with 1000 bootstrap replicates. The Criollo group was used as the main axis to display the tree.

### 4.4. Production Systems

#### 4.4.1. Experimental Areas, Plant Material, and Cultivation Conditions

Data were collected in three experimental areas on cacao-producing farms in the southern region of Bahia, Brazil. Each experimental area featured a distinct production system: E1—cacao cultivated in full sun with 100% Photosynthetically Active Radiation (PAR); E2—cacao cultivated in thin cabruca (80% PAR); and E3—cacao cultivated in dense cabruca (65% PAR). These experiments were established in 2015, and data collection was conducted in 2022 [31]. The clonal cultivar seedlings (CCN 51, FA 13, PS 1319, and SJ 02) were produced by lateral grafting above the cotyledon and below the first pair of leaves on open-pollinated seedling rootstocks of the clonal cultivar TSH 1188. Five months after grafting, the seedlings were transplanted to the field at a spacing of 2.60 m (between rows) and 3.00 m (between cacao trees), equivalent to a density of 1282 plants per hectare in a triangular arrangement over an area of 0.3 hectare. Banana plants were initially planted in the experimental areas for temporary shading, and subsequently, the new cacao seedlings were planted. Two years after planting, the banana plants were removed. Soil physicochemical analyses were conducted, and management and fertilization practices were carried out as described by de Souza Júnior [60].

#### 4.4.2. Potential Cacao Production

Data were collected on the number of healthy pods (pod plant^−1^) and productivity (kg plant^−1^) by counting the number of healthy pods produced monthly per plant in 2022. Only the production from the sixth to the seventh year in the field was used to analyze the clones’ potential productivity, as at this age, the plants reached physiological maturity and achieved production stability. To convert the mass of fresh seeds into productivity (kg of dry cocoa beans plant^−1^), pod indices from 2022 for each cultivar were used [31].

#### 4.4.3. Disease Incidence and Pod Damage

Counts of infected pods were collected in the same manner as healthy pods, collecting pods with symptoms of witch’s broom disease (*M. perniciosa*) and/or black pod rot (*Phytophthora* spp.), as well as damaged pods (mummified, damaged by animals, and other causes). Percentages and the annual average of infected and damaged pods relative to the total number of pods were calculated.

#### 4.4.4. Experimental Design

The three experiments were conducted in field conditions using a randomized block design. Each experiment (E1—cacao cultivated in full sun; E2—cacao cultivated in thin cabruca or less dense shade; E3—cacao cultivated in dense cabruca) consisted of four clonal cultivars (CCN 51, FA 13, PS 1319, and SJ 02), two blocks, and five useful plants per experimental unit. Statistical analyses were performed using data collected in 2022 for each experimental area and clonal cultivar.

### 4.5. Chemical, Physical, and Sensory Analysis

The liquor samples from the clonal cacao cultivars were obtained at the Cacausense Laboratory at the Cacao Innovation Center (CIC), Brazil, during the 2023 harvest year and sensorially analyzed using the mean values from the Quantitative Descriptive Analysis (QDA) following the CoEx protocol [61].

Physicochemical analyses of the cocoa beans were conducted by the Plant Classification Laboratory at CIC, following the international standard ISO 2451:2017 [62]. Chemical component analyses, including total polyphenols, purines, methylxanthines (caffeine and theobromine), and fat content, were conducted by Mérieux Nutrisciences, Italy.

### 4.6. Statistical Analysis

The experimental results of production systems were subjected to analysis of variance (ANOVA). Comparisons between means were made using the Tukey test. Analyses were conducted using the statistical software R version 4.1.3 [63].

Two principal component analyses (PCAs) were conducted, involving the selected clonal cultivars (BN 34, CCN 51, FA 13, PS 1319, and SJ 02) and Catongo. BN 34 and Catongo were used as benchmarks due to their status as award-winning sensory references of cacao in Brazil in recent years. The PCA analyses were performed separately for chemical and sensory attributes. For the PCA of organoleptic (sensory) attributes, the 11 variables (cocoa, bitter, acid, astringency, sweet, fresh fruits, brown fruits, nutty, floral, spice, and wood) were subjected to analysis of variance (ANOVA), factor analysis, and correlations. Six variables (cocoa, bitter, acid, sweet, fresh fruits, and spice) were used for the PCA due to their significant differences among the clonal cultivars, factor loadings ≥ 0.7, and lack of correlation or negative correlation.

## 5. Conclusions

The findings of this study demonstrate that Brazil’s elite cocoa clones, derived from complex genetic crosses primarily involving the Amelonado, Contamana, Iquitos, and Criollo groups, exhibit substantial productive potential, notable disease resistance, and desirable physico-chemical and organoleptic attributes. The genetic diversity observed among these clones highlights their adaptability and suitability under varied cultivation conditions, offering breeders valuable genetic resources for future improvement. Thus, incorporating these elite clones as parental lines in clonal breeding programs can expedite the development of new cocoa cultivars with optimized yield, improved resistance to prevalent diseases, and superior sensory qualities. Such advances have significant implications for enhancing the sustainability and global competitiveness of cocoa production in Brazil and other cacao-producing regions.

## Figures and Tables

**Figure 1 ijms-26-03386-f001:**
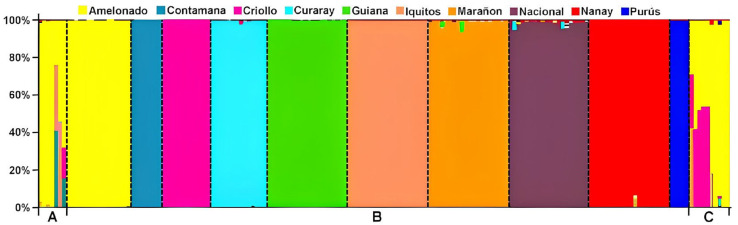
Ancestry of elite clonal cultivars, local cultivars, and reference accessions from the International Cocoa Genebank Trinidad (ICGT) analyzed in STRUCTURE Version 2.3.4 [22]. A—Seven Brazilian cultivars (Pará, Parazinho, Comum, Maranhão, SJ 02, FA 13, PS 1319, respectively). B—Reference accessions of the cocoa genetic groups described by Motamayor et al. [8]. C—Ten reference accessions selected from the ICGT (CCN 51, ICS 1, ICS 40, ICS 60, ICS 95, M 8, MATINA 1/6, REDAMEL 1/27, REDAMEL 1/30, REDAMEL 1/31).

**Figure 2 ijms-26-03386-f002:**
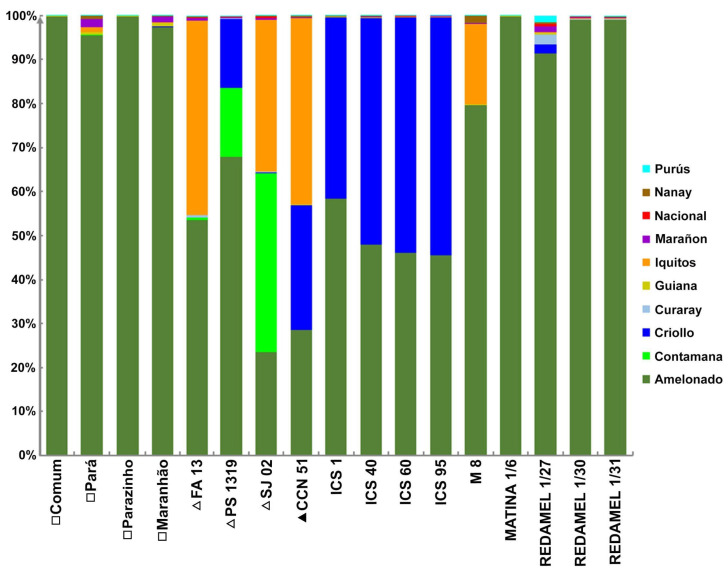
Genetic structure of local cultivars from Bahia (□), elite clonal cultivars from Bahia (Δ), clone CCN 51 (▲), and reference accessions from the International Cocoa Genebank Trinidad (ICGT), analyzed using STRUCTURE Version 2.3.4 [22]. Colors represent cocoa genetic groups previously described by Motamayor et al. [8].

**Figure 3 ijms-26-03386-f003:**
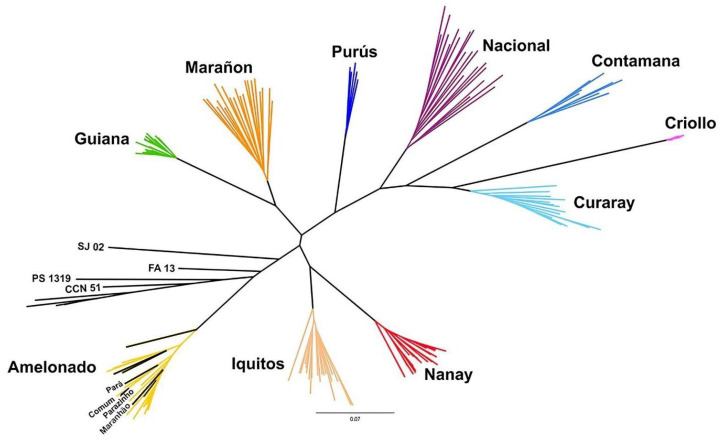
Phylogenetic tree constructed using the Neighbor Joining method based on 187 SNPs, with elite Brazilian clonal cultivars (SJ 02, FA 13, PS 1319), clone CCN 51, local cultivars from Bahia (Pará, Comum, Parazinho, and Maranhão), reference cultivars from the International Cocoa Genbank Trinidad, and the ten genetic groups described by Motamayor et al. [8].

**Figure 4 ijms-26-03386-f004:**
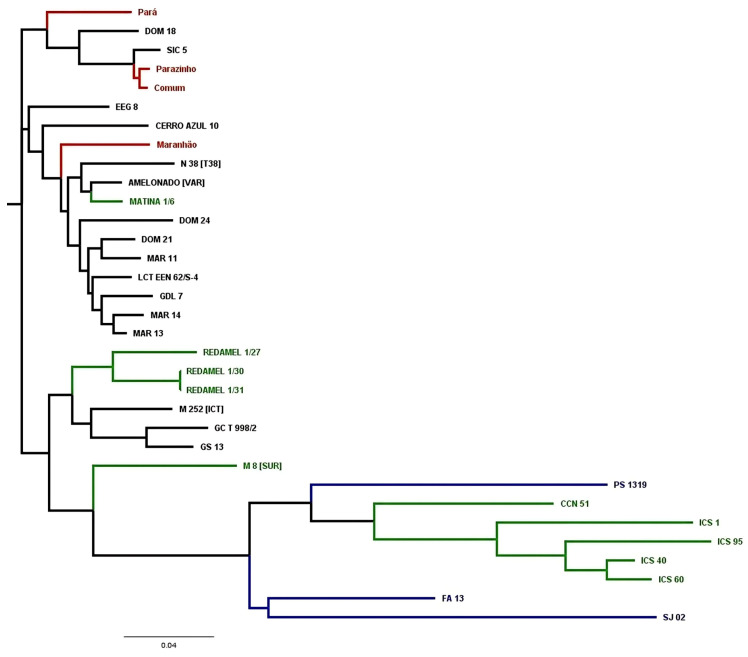
Dendrogram showing a close-up of the Amelonado group from Figure 3, with Brazilian clonal cultivars in blue, local cultivars from Bahia in red, CCN 51 and reference genotypes from the International Cocoa Genebank Trinidad in green, and reference cultivars of the Amelonado genetic group in black.

**Figure 5 ijms-26-03386-f005:**
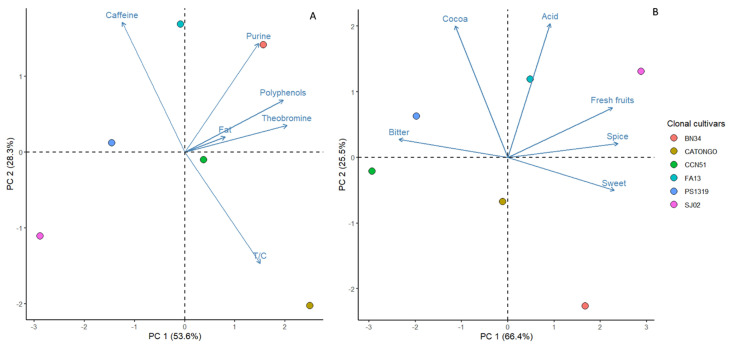
Principal components analysis (PCA) for the chemical (**A**) and sensory (**B**) attributes of the beans from six clonal cacao cultivars in Bahia, Brazil.

**Table 1 ijms-26-03386-t001:** Distribution of internationally defined genetic groups, previously described by Motamayor et al. [8], within the genetic structure of local and clonal cultivars from Bahia, Brazil.

	Local Cultivars				Clonal Cultivars		
Genetic Group	Comum	Pará	Parazinho	Maranhão	PS 1319	SJ 02	FA 13
Amelonado	0.9980	0.9550	0.9980	0.9740	0.6790	0.2350	0.5350
Contamana	0.0001	0.0010	0.0001	0.0004	0.1560	0.4070	0.0052
Criollo	0.0001	0.0004	0.0001	0.0004	0.1570	0.0007	0.0011
Curaray	0.0001	0.0005	0.0001	0.0008	0.0011	0.0021	0.0043
Guiana	0.0004	0.0057	0.0004	0.0077	0.0007	0.0013	0.0011
Iquitos	0.0002	0.0102	0.0002	0.0010	0.0012	0.3450	0.4410
Marañon	0.0005	0.0186	0.0005	0.0138	0.0017	0.0024	0.0064
Nacional	0.0000	0.0006	0.0001	0.0005	0.0009	0.0047	0.0013
Nanay	0.0000	0.0068	0.0004	0.0009	0.0009	0.0009	0.0015
Purús	0.0000	0.0013	0.0002	0.0004	0.0015	0.0010	0.0031

**Table 2 ijms-26-03386-t002:** Means and summary of the analysis of variance for the number of healthy pods’ plant^−1^ and productivity (kg plant^−1^) of four cacao clonal cultivars from three different production systems in Bahia, Brazil, in 2022.

Production System	Clonal Cultivar	Healthy Pods Plant^−1^		Dry Bean (kg Plant^−1^)	
Full sun	CCN 51	37.6	c	2.6	b
	FA 13	88.6	a	4.0	a
	PS 1319	54.4	bc	3.0	b
	SJ 02	68.2	ab	3.9	a
Thin cabruca	CCN 51	53.1	ab	3.7	a
	FA 13	71.3	ab	3.2	a
	PS 1319	85.5	a	4.7	a
	SJ 02	37.2	b	2.1	a
Dense cabruca	CCN 51	19.3	b	1.3	b
	FA 13	62.9	a	2.8	a
	PS 1319	22.7	b	1.2	b
	SJ 02	29.1	b	1.7	b
	Source of variation	F value	Pr (>F)	F value	Pr (>F)
Full sun	Clonal cultivar	30.61	0.009 **	10.35	0.043 *
	Block	0.11	0.761	0.01	0.938
Thin cabruca	Clonal cultivar	11.01	0.040 *	9.02	0.052
	Block	7.85	0.068	7.80	0.068
Dense cabruca	Clonal cultivar	35.02	0.008 **	19.30	0.018 *
	Block	17.63	0.025 *	21.90	0.018 *

* *p* < 0.05, ** *p* < 0.01. Lowercase letters indicate differences between clonal cultivars within each production system. Mean comparisons were performed using Tukey’s test (*p* < 0.05).

**Table 3 ijms-26-03386-t003:** Mean and summary of variance analysis of percentages of witches’ broom, black pod rot, and damage (mummified and others) in pods of four clonal cacao cultivars from three different production systems in Bahia, Brazil, in 2022.

Production System	Clonal Cultivar	Witches’ Broom		Black Pod Rot		Damage	
		(% Plant^−1^)					
Full sun	CCN 51	2.98		0.44		18.59	
	FA 13	7.26		0.29		8.68	
	PS 1319	8.11		2.86		10.74	
	SJ 02	7.91		1.84		13.13	
Thin cabruca	CCN 51	4.87		4.14		10.80	
	FA 13	4.34		0.66		14.19	
	PS 1319	2.38		1.50		7.27	
	SJ 02	4.34		0.55		11.18	
Dense cabruca	CCN 51	5.75		10.49		8.84	
	FA 13	6.26		3.21		11.47	
	PS 1319	7.38		7.58		16.44	
	SJ 02	7.62		2.01		13.74	
	Source of variation	F value	Pr (>F)	F value	Pr (>F)	F value	Pr (>F)
Full sun	Clonal cultivar	3.59	0.16	8.95	0.05	4.15	0.14
	Block	0.84	0.43	0.17	0.71	0.09	0.78
Thin cabruca	Clonal cultivar	0.40	0.76	2.58	0.23	0.57	0.67
	Block	1.64	0.29	1.76	0.28	0.09	0.79
Dense cabruca	Clonal cultivar	0.40	0.77	1.40	0.39	0.64	0.64
	Block	17.33	0.03 *	6.46	0.08	0.35	0.59

* *p* < 0.05. The results show the percentage relative to the total number of pods produced per plant in 2022.

**Table 4 ijms-26-03386-t004:** Beans’ physical, chemical, and sensory attributes from six clonal cacao cultivars in Bahia, Brazil.

Attributes	BN 34	Catongo	CCN 51	FA 13	PS 1319	SJ 02
Physical						
Beans (no./100 g)	61.00	96.00	65.00	104.00	76.00	81.00
Avg bean mass (g)	1.64	1.04	1.54	0.96	1.31	1.23
pH (a 25 °C)	5.27	5.18	4.92	5.12	5.20	5.07
Humidity of bean (%)	6.50	5.50	4.90	7.00	8.10	7.90
Chemical						
Total purine (g/100 g)	1.23	1.14	1.18	1.23	1.13	0.97
Caffeine (g/100 g)	0.19	0.07	0.15	0.24	0.20	0.19
Theobromine (g/100 g)	1.04	1.07	1.02	0.99	0.93	0.79
Fat (g/100 g)	56.52	55.34	54.26	55.13	52.61	55.51
Total Polyphenols (g/100 g)	3.19	2.69	2.03	2.26	1.64	1.03
T/C *	5.36	15.18	6.75	4.07	4.58	4.18
Organoleptic						
Cocoa	4.10	4.29	4.71	4.88	4.98	4.60
Bitter	4.60	4.67	5.17	4.79	5.02	4.50
Acid	4.60	5.08	4.92	5.17	4.96	5.54
Astringency	5.05	4.79	4.81	4.60	4.98	4.54
Sweet	0.67	0.21	0.04	0.29	0.08	0.67
Fresh Fruits	3.25	3.02	2.69	3.42	2.92	3.58
Brown Fruits	2.60	2.65	2.52	3.08	2.77	3.52
Nutty	2.46	2.67	2.75	2.58	2.58	2.62
Floral	3.66	2.71	2.56	2.46	2.27	2.96
Spice	2.09	1.65	1.25	1.98	1.54	2.27
Wood	2.71	2.65	2.60	2.54	2.31	2.46

* theobromine and caffeine ratio.

**Table 5 ijms-26-03386-t005:** Principal component analysis (PCA) values for chemical and organoleptic attributes of cocoa beans from six clonal cultivars in Bahia, Brazil.

Attributes	PC 1	PC 2
Chemical		
Purine	0.69	0.67
Caffeine	−0.58	0.80
Theobromine	0.96	0.17
Fat	0.38	0.09
Polyphenols	0.92	0.32
T/C *	0.71	−0.69
Eigenvalue	3.12	1.70
% Total variance	53.63	28.35
Cumulative %	53.63	81.98
Organoleptic		
Cocoa	−0.47	0.82
Bitter	−0.96	0.11
Acid	0.38	0.83
Sweet	0.94	−0.21
Fresh fruits	0.93	0.31
Spice	0.98	0.09
Eigenvalue	3.99	1.53
% Total variance	66.41	25.45
Cumulative %	66.41	91.86

* theobromine and caffeine ratio.

## Data Availability

The original contributions presented in the study are included in the article/Appendix A. Further inquiries can be directed to the corresponding author/s.

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
