# Peer review of "Elite Cacao Clonal Cultivars with Diverse Genetic Structure, High Potential of Production, and Good Organoleptic Quality Are Helping to Rebuild the Cocoa Industry in Brazil"

_ijms, 2025, doi:10.3390/ijms26073386_

Round 1

Reviewer 1 Report

Comments and Suggestions for Authors

This manuscript presents an experimental study on the clones like PS 1319, FA 13, and SJ 02 have been selected for their genetic structure, productivity, disease resistance, and physico-chemical and organoleptic characteristics. These elite clones, derived from complex crosses involving Amelonado, Contamana, Iquitos, and Criollo groups, are productive, resistant, and exhibit favorable physico-chemical and organoleptic qualities, making them valuable for future breeding programs. The reviewer appreciates the effort of the authors to prove their hypothesis using series of experiments. However, the reviewer has major suggestions regarding this manuscript. Thus, the authors need to consider the following comments to improve the quality of this manuscript.

Lines 88-92: Briefly improve it with specific objectives and scope of the study.

Lines 446-448: Authors have mentioned “Thousands of SNP loci… ten chromosomes of the cacao genome”. It would be useful to include the information on how these SNP loci were predicted and selected for the present study instead of stating the references.

In section 4.2 of materials and methods, please provide more details on the parameters used in the STRUCTURE software, including the reason for the choice of burn-in length, MCMC iterations and admixture model. While authors have provided some information in the results section, a more detailed explanation in section 4.2 would be helpful to the readers.

Lines 507, 508, 527, 528: Follow the uniformity of punctuation marks. Eg. 1000 replicates; 1,000 bootstrap replicates. Check and correct it.

The authors are asked to carefully check the formatting, punctuation, space errors, symbols, etc., in the entire manuscript.

Authors should provide a separate statistical analysis section in the materials and methods part. 

In section 2.3: The authors are advised to improve the disease resistance data. Though the authors mentioned that there were no significant differences in disease resistance levels among the cultivars, the Table 3 data revealed that there were some differences in the % of witches' broom and black pod rot. Additional discussion of these findings, including other reasons for the observed differences, would be helpful to the readers.

Enhance the resolution of figure 4.

In section 3.2: Please explain more about the pod index and yield differences among the cultivars.

If possible, please discuss a few lines about the importance of post-harvest processing in managing unfavorable characteristics in Section 3.3.

Provide a separate conclusion section and write a few lines about the limitations of the study and future directions or hypotheses about the study. It will be useful to the readers for ease of understanding to design their study related to this studied issue.

Please confirm that the entire manuscript is thoroughly proofread to remove any typo-errors.

Author Response

Reviewer 1

Reviewer 1. This manuscript presents an experimental study on the clones like PS 1319, FA 13, and SJ 02 have been selected for their genetic structure, productivity, disease resistance, and physico-chemical and organoleptic characteristics. These elite clones, derived from complex crosses involving Amelonado, Contamana, Iquitos, and Criollo groups, are productive, resistant, and exhibit favorable physico-chemical and organoleptic qualities, making them valuable for future breeding programs. The reviewer appreciates the effort of the authors to prove their hypothesis using series of experiments. However, the reviewer has major suggestions regarding this manuscript. Thus, the authors need to consider the following comments to improve the quality of this manuscript.

Authors’ Response. We appreciate the valuable considerations and suggestions provided through the detailed analysis of our manuscript, and we are committed to adopting all recommendations applicable to the subject matter by incorporating the pertinent changes to enhance the clarity, scientific rigor, and relevance of the results presented.

Reviewer 1. Lines 88-92: Briefly improve it with specific objectives and scope of the study.

Authors’ Response. We have modified to improve the objectives and scope.

Reviewer 1. Lines 446-448: Authors have mentioned “Thousands of SNP loci… ten chromosomes of the cacao genome”. It would be useful to include the information on how these SNP loci were predicted and selected for the present study instead of stating the references.

Authors’ Response. The information was included as suggested. 

Reviewer 1. In section 4.2 of materials and methods, please provide more details on the parameters used in the STRUCTURE software, including the reason for the choice of burn-in length, MCMC iterations and admixture model. While authors have provided some information in the results section, a more detailed explanation in section 4.2 would be helpful to the readers.

Authors’ Response. We added a brief explanation of the reason for the choice of each parameters.

Reviewer 1. Lines 507, 508, 527, 528: Follow the uniformity of punctuation marks. Eg. 1000 replicates; 1,000 bootstrap replicates. Check and correct it.

Authors’ Response. The punctuation marks were uniformized.

Reviewer 1. The authors are asked to carefully check the formatting, punctuation, space errors, symbols, etc., in the entire manuscript.

Authors’ Response. The entire manuscript was carefully checked.

Reviewer 1. Authors should provide a separate statistical analysis section in the materials and methods part. 

Authors’ Response. We have incorporated separately the statistical analysis section into the materials and methods part, as requested.

Reviewer 1. In section 2.3: The authors are advised to improve the disease resistance data. Though the authors mentioned that there were no significant differences in disease resistance levels among the cultivars, the Table 3 data revealed that there were some differences in the % of witches' broom and black pod rot. Additional discussion of these findings, including other reasons for the observed differences, would be helpful to the readers.

Authors’ Response. In the analysis of variance for the percentages of witches' broom and black pod rot, the significance level was set at p < 0.05. Thus, significant differences were observed only among blocks for % witches' broom in the cabruca system.

Reviewer 1. Enhance the resolution of figure 4.

Authors’ Response. The resolution of figure 4 was improved. 

Reviewer 1. In section 3.2: Please explain more about the pod index and yield differences among the cultivars.

Authors’ Response. Our discussion was focused on a general explanation about production potential. Other papers will explore in details production components.

Reviewer 1. If possible, please discuss a few lines about the importance of post-harvest processing in managing unfavorable characteristics in Section 3.3.

Authors’ Response. The importance of post-harvest processing was included in the text.

Reviewer 1. Provide a separate conclusion section and write a few lines about the limitations of the study and future directions or hypotheses about the study. It will be useful to the readers for ease of understanding to design their study related to this studied issue.

Authors’ Response. The conclusion section was done.

Reviewer 1. Please confirm that the entire manuscript is thoroughly proofread to remove any typo-errors.

Authors’ Response. The entire manuscript was revised to remove any typo-errors.

Reviewer 2 Report

Comments and Suggestions for Authors

The manuscript “Elite cacao clonal cultivars with diverse genetic structure, high potential of production, and good organoleptic quality are helping to rebuild the cocoa industry in Brazil” is an interesting, original and integrative contribution to understand the importance of diferent studies in clonal cultivars characterization. Potentially the findings of the work, that indicates that the elite clones, derived from complex crosses, are productive, resistant, and exhibit favorable physicochemical and organoleptic qualities, can be very useful in future breeding programs. 

"Subject and importance of the work:

Manuscript presented for review with title: “Elite cacao clonal cultivars with diverse genetic structure, high potential of production, and good organoleptic quality are helping to rebuild the cocoa industry in Brazil” is well written and well discussed. The work is interesting and very important for the study field. Is an original and integrative contribution to understand the importance of the studies of the genetic ancestry, phylogeny, potential production, resistance, organoleptic, and physicochemical qualities of clonal cultivars, in their characterization.

Potentially the findings of the work, that indicates that the elite clones, derived from complex crosses, are productive, resistant, and exhibit favorable physicochemical and organoleptic qualities, can be very useful in future breeding programs. This work highlights the importance of multidisciplinary approaches in cacao research, integrating several areas namely genetics, chemistry and sensory characteristics, to develop high-quality products that can be used by industry.

Manuscript

The title is adequate to their text, perhaps too long, try to shorten it

The experiment was planned very carefully.

Abstract: This manuscript part is well organized.

Keywords are adequate to manuscript text.

The Introduction section includes all necessary information about studied matters and problems.

Material and methods are adequately described

Results: All Tables and Figures are constructed well and contain all necessary information.

The discussion is clear and appropriate for the results obtained and considering that there are not many studies in this area, in this species, with these molecular markers. Is clear and in accordance with results.

There are no conclusions, they should add this section.

The references are appropriate for this work.

Global: The presented manuscript is very valuable and could be published in the International Journal of Molecular Sciences.

This work is totally integrated in the Special Issue “Plant Biodiversity and Molecular Marker Technology: Discovery and Application of DNA Polymorphisms”."

Author Response

Reviewer 2

Reviewer 2. The manuscript “Elite cacao clonal cultivars with diverse genetic structure, high potential of production, and good organoleptic quality are helping to rebuild the cocoa industry in Brazil” is an interesting, original and integrative contribution to understand the importance of different studies in clonal cultivars characterization. Potentially the findings of the work, that indicates that the elite clones, derived from complex crosses, are productive, resistant, and exhibit favorable physicochemical and organoleptic qualities, can be very useful in future breeding programs.

Reviewer 2. Manuscript presented for review with title: “Elite cacao clonal cultivars with diverse genetic structure, high potential of production, and good organoleptic quality are helping to rebuild the cocoa industry in Brazil” is well written and well discussed. The work is interesting and very important for the study field. Is an original and integrative contribution to understand the importance of the studies of the genetic ancestry, phylogeny, potential production, resistance, organoleptic, and physicochemical qualities of clonal cultivars, in their characterization.

Reviewer 2. Potentially the findings of the work, that indicates that the elite clones, derived from complex crosses, are productive, resistant, and exhibit favorable physicochemical and organoleptic qualities, can be very useful in future breeding programs. This work highlights the importance of multidisciplinary approaches in cacao research, integrating several areas namely genetics, chemistry and sensory characteristics, to develop high-quality products that can be used by industry.

Authors’ Response: Thank you very much for taking the time to review this manuscript. All corrections were made, as suggested.

Reviewer 2. The title is adequate to their text, perhaps too long, try to shorten it

Authors’ Response: Although it is longer than the average title, we chose to keep the original title because it reflects the breadth and complexity of the study demonstrating the impact of these cultivars on the reconstruction of the cocoa industry in Brazil. Shortening it would risk omitting crucial information about the robustness and multidimensional impact of the research.

Reviewer 2. The experiment was planned very carefully.

Reviewer 2. Abstract: This manuscript part is well organized.

Reviewer 2. Keywords are adequate to manuscript text.

Reviewer 2. The Introduction section includes all necessary information about studied matters and problems.

Reviewer 2. Material and methods are adequately described

Reviewer 2. Results: All Tables and Figures are constructed well and contain all necessary information.

Reviewer 2. The discussion is clear and appropriate for the results obtained and considering that there are not many studies in this area, in this species, with these molecular markers. Is clear and in accordance with results.

Reviewer 2. There are no conclusions, they should add this section.

Authors’ Response: We added the conclusion into the text.

Reviewer 2. The references are appropriate for this work.

Reviewer 2. Global: The presented manuscript is very valuable and could be published in the International Journal of Molecular Sciences.

Reviewer 2. This work is totally integrated in the Special Issue “Plant Biodiversity and Molecular Marker Technology: Discovery and Application of DNA Polymorphisms”."

Reviewer 3 Report

Comments and Suggestions for Authors

It is a good work, which deserves to be published, but I felt some additions were necessary. In particular, with regard to clone production, I think it would be useful to have an idea of the various characteristics that explain the differences in production (apart from diseases): self-compatibility, number of pods, number of beans per pod, average weight of a bean. I think it's important to detail the components of production.

Author Response

Reviewer 3

Reviewer 3. It is a good work, which deserves to be published, but I felt some additions were necessary. In particular, with regard to clone production, I think it would be useful to have an idea of the various characteristics that explain the differences in production (apart from diseases): self-compatibility, number of pods, number of beans per pod, average weight of a bean. I think it's important to detail the components of production.

Authors’ Response: We appreciate the reviewer’s valuable suggestion. However, our objective was to provide a general characterization of the evaluated genotypes, emphasizing their high production potential and distinct genetic profiles. The observed higher productivity is linked to the genetic structure and ancestry of the materials and their acclimation capacity to the environmental conditions studied. A detailed analysis of production components was not within the scope of the current manuscript, as explicitly stated in the study objectives. Such detailed analyses will be thoroughly addressed in future studies, which will involve a greater number of clones.

Round 2

Reviewer 1 Report

Comments and Suggestions for Authors

The authors have suitably incorporated my comments in the revised manuscript. Now, the overall quality of the manuscript has significantly improved. Therefore, I recommend the manuscript be accepted for publication in its current form.